# Structure and Catalytic Performance of Carbon-Based Solid Acids from Biomass Activated by ZnCl₂

Yao Wu [1], Hao Zhang [1], Zhaozhou Wei [1], Deyuan Xiong [1,2,*] ID, Songbai Bai [3], Menglong Tong [1] and Pengcheng Ma [1]

1 College of Chemistry and Chemical Engineering, Guangxi University, Nanning 530004, China; 2014302069@st.gxu.edu.cn (Y.W.); zhwjszs@163.com (H.Z.); zzwei0608@outlook.com (Z.W.)
2 Guangxi Key Laboratory of Petrochemical Resource Processing and Process Intensification Technology, School of Chemistry and Chemical Engineering, Guangxi University, Nanning 530004, China
3 College of Foreign Languages, Guangxi University, Nanning 530004, China; baisb16@lzu.edu.cn
* Correspondence: dyxiong@gxu.edu.cn

**Abstract:** In the current investigation, carbon-based solid acid catalysts were synthesized from peanut shells (PSs) and rice straw (RS) using ZnCl₂ activation and concentrated sulfuric acid sulfonation. These catalysts were then employed for the hydration of pinene to produce terpineol. The research findings suggest that the natural porous structure of RS is more amenable to ZnCl₂ activation compared to PSs. Furthermore, the catalysts prepared from fully activated RS by ZnCl₂ (RSA-C-S) had a higher $S_{BET}$ and higher density of oxygen-containing groups (–COOH) in comparison with unactivated RS-based solid acids (RSC-S). The characterization outcomes revealed that RSA-C-S possesses a specific surface area of 527.0 m²/g, significantly outperforming RSC-S, which has a surface area of 420.9 m²/g. Additionally, RSA-C-S registered a higher –COOH density of 1.37 mmol/g, as opposed to RSC-S's, with 1.07 mmol/g, attributable to the partial oxidation of internal –OH groups during activation. Experimental data from hydration tests confirmed that the catalyst's superior performance is largely attributed to its elevated specific surface area and a high density of –COOH functional groups. Under optimal reaction parameters, RSA-C-S demonstrated unparalleled catalytic efficiency in the synthesis of α-terpineol via hydration of α-pinene, achieving conversion and selectivity rates of 87.15% and 54.19%, respectively.

**Keywords:** solid acid catalyst; biomass; rice straw; chemical activation

## 1. Introduction

α-terpineol, characterized by its chemical formula C₁₀H₁₈O, is a monoterpenoid alcohol naturally present in various plant species. Nonetheless, the synthetic production of α-terpineol has been established for over a century, finding extensive application in the fragrance sector. Serving as a key raw material, α-terpineol contributes to various perfumery products and is globally produced at a scale of thousands of tons each year [1]. α-terpineol is widely used in industry, agriculture [2], and medicine. Concurrently, α-terpineol acts as a fundamental raw material in the chemical industry for the production of flotation minerals, cleaning agents, and disinfectants [3]. The chemical synthesis of α-terpineol serves not only to augment the commercial value of turpentine but also to foster economic development in the forestry sector. Industrially, α-terpineol is predominantly synthesized through a "two-step method" involving 30% diluted H₂SO₄. This method initially catalyzes turpentine to form terpin hydrate, followed by its dehydration using dilute acid to yield α-terpineol.

Nonetheless, the use of liquid acid as a catalyst presents significant challenges, including product separation difficulties, equipment corrosion, and environmental contamination. A viable alternative to address these issues is the utilization of solid acid catalysts. Prevalent heterogeneous solid acid catalysts include zeolites, ion-exchange resins, molecular sieves, and membrane materials [4–8]. Contemporary research focusing on the catalytic synthesis

of α-terpineol predominantly employs a "one-step method" operating within a heterogeneous system. Vital et al. [6] used USY zeolite, β zeolite, or a surface-modified activated carbon polydimethylsiloxane membrane (PDMS) catalyst for the hydration reaction of α-pinene to yield a maximum selectivity of 70.00%. Especially for the USY zeolite, the increase in membrane channeling leads to an increase in reagent permeation, further improving the catalytic activity. Yang et al. [5] used Amberlyst 15 wet acid as the catalyst in the hydration of α-pinene in a pilot-scale jet reactor, wherein the conversion of α-pinene and selectivity of α-terpineol reached 85.5% and 43.0%, respectively. The Amberlyst 15 wet catalyst has good catalytic activity because of its specific surface area. A large pore size provides a good physical adsorption capacity and promotes the hydration reaction of α-pinene. While traditional solid acid catalysts offer benefits such as a high acidity and extensive specific surface area, they are also burdened by shortcomings, including elevated preparation costs, intricate fabrication processes, limited catalytic activity, and the instability of active sites. Moreover, the hydration of α-pinene results in the formation of numerous by-products, as depicted in Scheme 1. Consequently, it is necessary to find an effective catalyst to replace the traditional solid acid catalyst.

**Scheme 1.** Reaction pathways for the hydration and isomerization of α-pinene.

Researchers have increasingly advocated for biomass carbon-based catalysts due to their well-defined structural morphology, high catalytic activity [9], sustainability, and cost-effectiveness. Empirical evidence suggests that employing biomass as a substrate for catalyst preparation confers merits such as robust catalytic activity, stability, and recyclability [10–14]. As a result, these catalysts have emerged as promising solid acid alternatives, finding applications in various chemical reactions, including esterification processes. Chen et al. [15], Thushari et al. [11], and Mardhiah et al. [16] prepared biomass carbon-based solid acid catalysts by the carbonization-sulfonation two-step process, which has been used to synthesize biodiesel. Previous investigations have demonstrated that the carbon backbone structure is contingent upon both the intrinsic material composition and the temperature conditions during carbonization. Incomplete carbonization yields a "soft" carbon framework that is amenable to functionalization with catalytically active acidic groups, thereby enhancing the performance metrics of the biomass carbon-based solid acid catalysts.

While biomass-derived solid acids address the issues associated with liquid acids, such as poor recoverability and equipment corrosion [17], there remains room for optimizing the pore structure of these biomass materials. Such optimization can enhance pore utilization, and this can be achieved through chemical activation methods; specifically, by using agents like $ZnCl_2$ [18]. The chemical activation mechanism employs $ZnCl_2$ etching and high-temperature gas generation to modify the porous structure of biomass-derived solid acids. This approach serves to either create additional pores on the external surface or regulate the interconnectivity between internal and external pores. As a result, the porosity of the biomass carbon-based solid acids is effectively increased, facilitating a greater ac-

commodation of active centers. This, in turn, provides a foundation for enhanced catalytic performance and reduced solid–liquid mass transfer resistance. Ma et al. [19] prepared microporous lignin-derived carbon-based solid acids (MLC-S) for the esterification of oleic acid with methanol by chemical activation with $ZnCl_2$ and sulfonation with concentrated sulfuric acid. Their results showed that MLC-S exhibits good catalytic performance and stability in the esterification reaction. Hussein et al. [20] used biomass waste potato peels (PPs) as a feedstock to produce carbon-based solid acid catalysts for biodiesel production. PPs were mixed with a $ZnCl_2$ solution and carbonized at 450 °C for 1 h to obtain porous carbon material (PPAC), which was then sulfonated by concentrated sulphuric acid to obtain a catalyst that exhibited excellent catalytic performance and good thermal stability in the acid-catalyzed esterification of oleic acid with methanol for biodiesel production. Xie et al. [21] prepared biomass carbon-based solid acid catalysts from sulfated lignin by a phosphoric acid pretreatment and incomplete carbonization (400 °C) that were loaded with –$SO_3H$ groups at 180 °C. The conversion of $\alpha$-pinene and the selectivity of $\alpha$-terpineol reached 95.3% and 55.3%, respectively. During the carbonization of lignin, the pore structure of the catalyst can be controlled by adjusting the amount of phosphoric acid, and the higher average pore size (6.73 nm) promotes the hydration of $\alpha$-pinene within the catalyst. However, the purification cost of lignin is high, and its sulfonation temperature is also high, which is inconsistent with economic benefits. Hence, there is a considerable risk in the preparation of lignin.

In this study, two mesoporous biomass carbon-based solid acid catalysts were prepared using $ZnCl_2$ activation, low-temperature carbonization (300 °C), and sulfonation (80 °C) for the preparation of $\alpha$-terpineol by the $\alpha$-pinene hydration reaction. The activation carbonization method first prepared the biomass carbon precursors PSA-C and RSA-C using two agricultural wastes—peanut shells (PSs) and rice straw (RS)—and $ZnCl_2$ as chemical activators. Then, PSA-C-S and RSA-C-S were prepared by sulfonating the PSA-C and RSA-C with concentrated sulfuric acid. Finally, we characterized the chemical and physical properties of the carbon precursors and catalysts. The study investigated the influence of $ZnCl_2$ activation on both the specific surface area and the distribution of acidic oxygen-containing functional groups within the two types of carbon precursors and catalysts. Employing the hydration reaction of $\alpha$-pinene to synthesize $\alpha$-terpineol as the target reaction, the research further examined the effects of the catalyst-specific surface area and internal acidic oxygen-containing functional group distribution on catalytic performance. Additionally, the optimal hydration reaction conditions under catalyst usage were identified. Given the plentiful availability of raw materials such as peanut shells and rice straw, coupled with the simplicity of the preparation procedure, the experimental methodology introduced in the current study offers both environmental sustainability and economic viability.

## 2. Results and Discussion

### 2.1. Characterization of Carbon-Based Solid Acid Catalysts

#### 2.1.1. Thermal Stability

The thermal stability of both rice straw and peanut shells was assessed using a synchronous thermal analyzer. Figure 1 illustrates that the weight loss of these two biomass materials can be categorized into three primary phases. The initial phase of weight loss, occurring between 30 and 150 °C, is attributable to the evaporation of the water content within the biomass. Within the temperature range of 240–600 °C, both rice straw and peanut shells experience significant weight loss. This reduction is attributed to the degradation of hemicellulose, cellulose, and certain lignin components within the biomass, as well as to the cleavage of glycosidic bonds and the ring-opening of D-glucose. Concurrently, processes such as cyclization, aromatization, and cross-linking generate graphite-like aromatic carbon sheet structures, as evidenced by references [11,22,23]. This phase is also marked by a substantial release of carbon dioxide and carbon monoxide. At this stage, the loss of rice straw is greater than that of peanut shells, which can make rice straw more easily

activated by $ZnCl_2$ to produce more pores, thereby increasing the specific surface area. The terminal phase of weight loss transpires between 600 and 1000 °C, characterized by a slower degradation rate of the residual lignin in both types of biomasses, resulting in reduced weight loss. For the catalyst preparation, a carbonization temperature of 300 °C was selected with the intent to preserve the native structures of peanut shells. In contrast, carbonization at elevated temperatures has the potential to induce rigid structures [24], thereby compromising the integrity of the peanut shell architecture and adversely affecting subsequent functionalization.

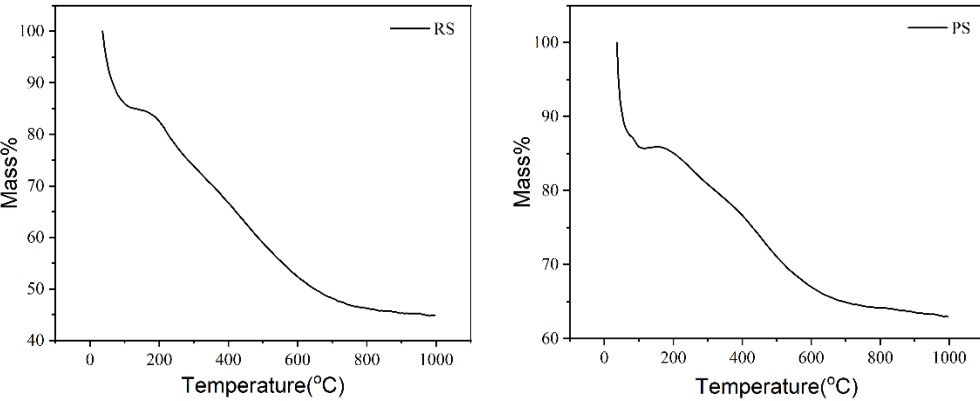

**Figure 1.** Thermogravimetric (TG) curves of rice straw and peanut shells.

### 2.1.2. The Structure of Pore and Surface Morphology

The peanut shell and rice straw carbon precursors and catalyst's pore structure and morphology were determined by SEM images, $N_2$-physical adsorption, desorption isotherms [25], and an aperture distribution diagram.

Figure 2 displays the SEM images of both the peanut shell and rice straw carbon precursors, as well as the catalysts. Figures 2a and 2b depict the peanut shell carbon precursors PSA and PSA-300, respectively. Examination reveals that the surface of PSA is smoother after the activation treatment and the porosity is not well improved. The carbonized PSA-300 maintains the structural integrity of the original biomass carbon, signifying the good thermal stability of the peanut shell. Figure 2c illustrates catalyst PSA-300-80 subsequent to sulfonation treatment using $H_2SO_4$. When contrasted with its carbon precursor, the catalyst exhibits a nuanced shift in surface morphology. Specifically, the structure manifests as a folded block featuring partial channel-like attributes. This results in an irregular surface accompanied by enhanced porosity. The disparity between these two structural morphologies can be ascribed to complex reactions, notably acid hydrolysis, occurring during the sulfonation process of the latter, leading to enhanced porosity [26]. Moreover, the addition of $-SO_3H$ groups to the carbon matrix results in a minor elevation of surface roughness, attributed to the uneven distribution of carbon particles. This surface irregularity is potentially mediated by hydrogen bonding between specific carbon particles and their associated $-SO_3H$ groups, causing them to aggregate [27]. Figure 2d–f present scanning electron microscopy images of the rice straw precursors RSA, RSA-300, and catalyst RSA-300-80, respectively. Upon examination, a proliferation of small pores is evident on the internal surface of the rice straw subsequent to $ZnCl_2$ activation, in contrast to peanut shells. This observation substantiates the efficacy of $ZnCl_2$ chemical activation in enhancing the porosity of rice straw. Concurrently, catalyst RSA-300-80 displays a complex pore architecture both on its surface and within its internal structure, signifying that $H_2SO_4$ sulfonation further augments the porosity of rice straw.

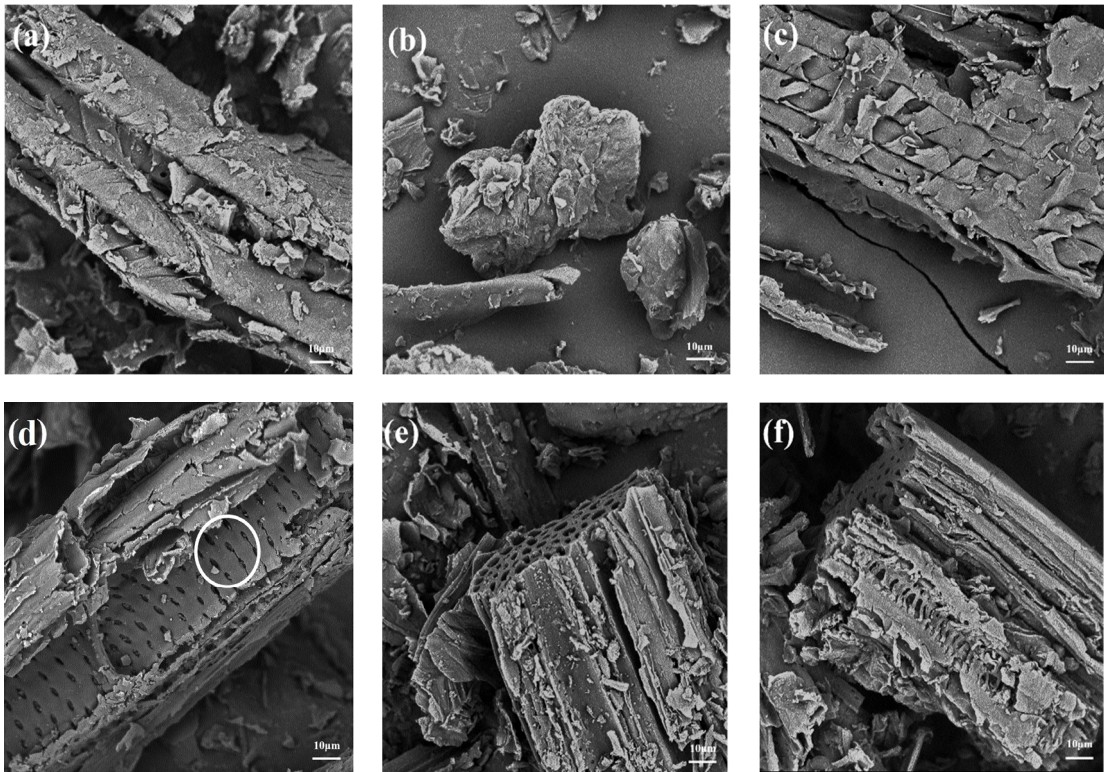

**Figure 2.** SEM images of (**a**) PSA; (**b**) PSA-300; (**c**) PSA-300-80; (**d**) RSA; (**e**) RSA-300; and (**f**) RSA-300-80.

Figure 3 illustrates the pore size distribution and $N_2$ physical adsorption–desorption isotherms for both carbon precursors and catalysts derived from peanut shells and rice straw. Upon analysis, it is evident that the pore sizes predominantly range between 2 and 5 nm, with a negligible presence of micropores. These findings categorize the synthesized catalyst as primarily mesoporous material [28]. As the relative pressure escalates, the adsorption and desorption isotherms for all biomass-based carbon precursors and catalysts exhibit analogous hysteresis loops. These isotherms conform to Type IV as defined by the International Union for Pure and Applied Chemistry (IUPAC), further corroborating the mesoporous nature of the prepared materials.

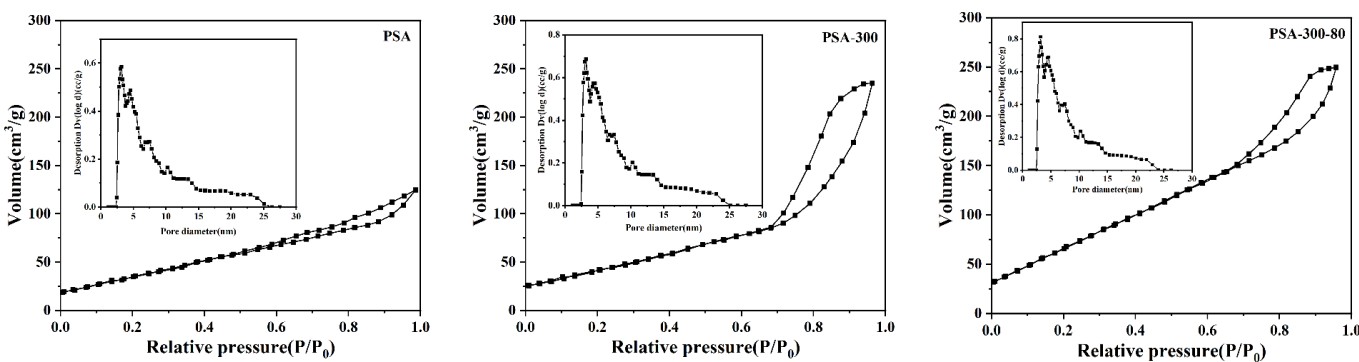

**Figure 3.** *Cont.*

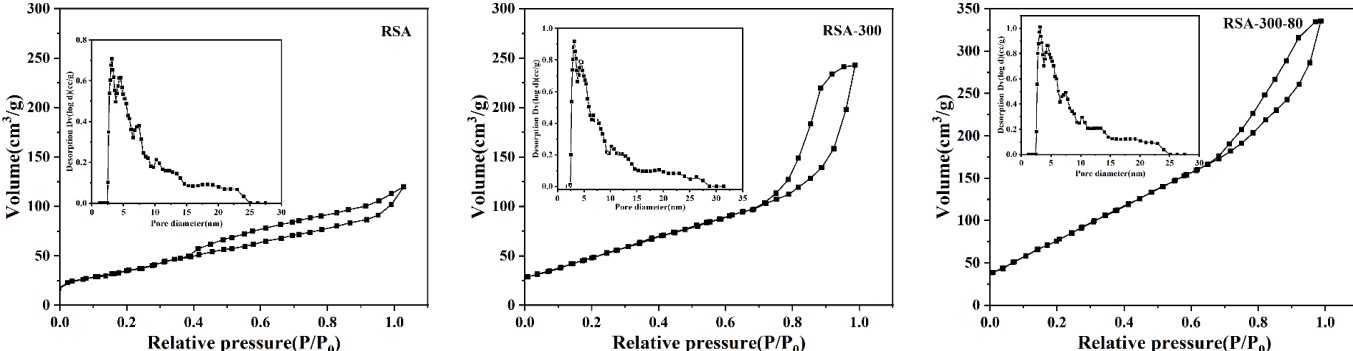

**Figure 3.** Pore size distribution and $N_2$-adsorption desorption isotherms of carbon precursors and catalysts.

Table 1 presents the $N_2$ physisorption results for the biomass-derived carbon precursors and catalysts. Both the specific surface area and total pore volume exhibit an increase following the carbonization and sulfonation processes. Notably, the pore dimensions of the rice straw-derived materials surpass those of the peanut shell derivatives. This enhancement in the specific surface area post-sulfonation is attributable to the dual role of concentrated sulfuric acid, which serves as both a sulfonation agent and an oxidant. This dual functionality results in the degradation of the original, thinner pore walls and the subsequent formation of new porous architectures [29]. These findings underscore the efficacy of $ZnCl_2$ activation in augmenting pore structure, particularly for biomass sharing structural similarities with rice straw. The specific surface area of the peanut shells increased from 306.3 $m^2/g$ (carbon precursor) to 425.9 $m^2/g$ (catalyst), and the total pore volume increased from 0.28 $cm^3/g$ (carbon precursor) to 0.39 $cm^3/g$ (catalyst). The specific surface area of rice straw increased from 378.9 $m^2/g$ (carbon precursor) to 527.0 $m^2/g$ (catalyst), and the total pore volume increased from 0.35 $cm^3/g$ (carbon precursor) to 0.49 $cm^3/g$ (catalyst). For comparison, Yarusova et al. prepared a series of sodium aluminosilicate adsorbents using rice straw as raw material to adsorb methylene blue with a specific surface area of only 364 $m^2/g$. Alterations in the biomass carbon structure, such as the collapse and fracturing of carbon frameworks, were evident following carbonization and sulfonation [30]. During catalyst preparation, the cleavage of glycosidic bonds in cellulose and hemicellulose leads to localized disruptions in the pore structure of the carbon precursor. Consequently, these disruptions contribute to the elevated porosity in the resultant catalyst.

**Table 1.** Properties of carbon precursors and catalysts.

| Samples | $S_{BET}$ ($m^2/g$) | $V_{Total}$ ($cm^3/g$) | $D_{pore}$ (nm) |
|---|---|---|---|
| PS | 365.1 | 0.33 | 3.69 |
| PSA | 306.3 | 0.28 | 3.63 |
| PSA-300 | 362.3 | 0.34 | 3.72 |
| PSA-300-80 | 425.9 | 0.39 | 3.63 |
| PS300-80 | 490.1 | 0.46 | 3.79 |
| RS | 390.6 | 0.39 | 3.56 |
| RSA | 378.9 | 0.35 | 3.66 |
| RSA-300 | 457.7 | 0.46 | 3.98 |
| RSA-300-80 | 527.0 | 0.49 | 3.73 |
| RS300-80 | 420.9 | 0.41 | 3.84 |

Following $ZnCl_2$ activation, the specific surface area of carbon precursors PSA and RSA experiences a reduction due to pore obstruction by the residual zinc chloride in both peanut shells and rice straw. Subsequent carbonization and acid-washing processes effectively remove these obstructions. Notably, the specific surface area of RSA-300-80 exhibits an increase from 420.9 $m^2/g$ to 527.0 $m^2/g$ when compared to its non-activated counterpart,

RS300-80 (where 300 and 80, respectively, denote the carbonization and sulfonation temperatures, the same as the following). $ZnCl_2$ activation enhances the thermal sensitivity of the biomass, transforming the drying process in an electric blast oven into not only a dehydration step but also a preliminary carbonization phase. This pre-carbonization effectively removes volatile components and initiates pore formation within the biomass. Concurrently, $ZnCl_2$ contributes to lignocellulose dehydrogenation during activation, reducing tar formation and contributing to an increased specific surface area in RSA-300-80 [31,32]. In contrast, PSA-300-80 exhibits an inverse trend compared to its non-activated counterpart, PS300-80 (where 300 and 80 signify the carbonization and sulfonation temperatures, respectively, the same as the following). This divergence may be attributed to the softer texture of $ZnCl_2$-activated peanut shells, leading to partial pore collapse during sulfonation and, consequently, a reduction in the specific surface area [33]. Scanning electron microscopy images corroborate these observations.

### 2.1.3. Catalytic Active Center and Acid Density

Figure 4 presents the Fourier-transform infrared spectra, highlighting distinctions between the carbon precursors and catalysts derived from peanut shells and rice straw. Both sets of biomass carbon precursors and catalysts exhibit similar absorption bands in the primary regions centered around 1108, 1411, 1559, 1680, 1712, and 3080–3600 $cm^{-1}$. Notably, the catalysts display additional bands, absent in the carbon precursors, within the 600–900 $cm^{-1}$ range. This suggests varying degrees of C–H substitution in the catalysts compared to their precursor forms.

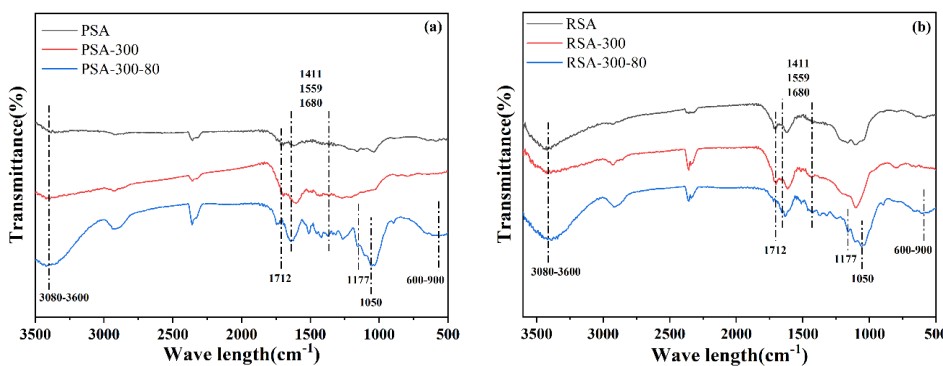

**Figure 4.** Fourier transform infrared spectroscopy (FT−IR) of (**a**) PS and (**b**) RS carbon precursors and catalysts.

Both the carbon precursor and catalyst spectra exhibit identical peaks around 1108 $cm^{-1}$ which are attributed to the C–O–C bonds in cyclic ethers such as D-glucose and glycosidic bonds. The bands appearing at 1411, 1559, and 1680 $cm^{-1}$ correspond to the C=C stretching vibrations in the aromatic skeletons. The 1712 $cm^{-1}$ band is indicative of the stretching vibrations of the carbon–oxygen double bonds in –C=O groups. Finally, the bands within the 3080–600 $cm^{-1}$ range are attributed to adsorbed water as well as alcoholic and phenolic –OH groups [15,34].

Distinct from the carbon precursor, the catalyst exhibits absorption bands in the 600 to 900 $cm^{-1}$ range, indicative of varying degrees of C–H bending and stretching [27,35]. This observation suggests that the catalyst likely has a higher degree of substitution than the carbon precursor, attributable to the inclusion of $–SO_3H$, –COOH, and –OH functional groups. Further, the presence of bending bands in the catalyst implies that the aliphatic C–H configurations are predominantly in the form of $–CH_2–$. The catalyst primarily contains cyclic aliphatic structures. Contrary to the carbon precursors, the catalyst exhibits a vibrational band around 1050 $cm^{-1}$, attributable to asymmetric O=S=O stretching. This feature is mainly associated with the extension of functional groups such as S=O or C–S. This phenomenon results from the successful incorporation of acidic $–SO_3H$ groups into the carbon structure [13,35]. Furthermore, an amplified peak intensity at 1712 $cm^{-1}$ is observed

in both PSA-300 and RSA-300-activated biomass. This intensification is likely attributable to the oxidation of intramolecular –OH groups to –COOH during the activation stage.

As depicted in Figure 5, the X-ray photoelectron spectroscopy spectra of both biomass carbon precursors and catalysts reveal peaks corresponding to binding energies of 168, 284, and 530 eV for S 2p, C 1s, and O 1s, respectively. Specifically, the S 2p diffraction peak at 167.9 eV is attributed to the –SO₃H sulfonic acid group in the catalyst. This suggests that the sulfur within the catalyst predominantly exists in the form of –SO₃H, while the peak at 169.5 eV is ascribed to sulfate [36]. The deconvolution of the C 1s peak uncovers the presence of aromatic carbon (C=C, 284.6 eV), hydroxyl (–C=O, 286.2 eV), carboxylic acid carbon (–COOH, 288.6 eV), and hemiacetal or carboxylic acid carbon (–C=O, 286.5–287.5 eV). Additionally, peaks corresponding to C=O and –O–C oxygen are identified at 531.6 and 532.5 eV, respectively [37].

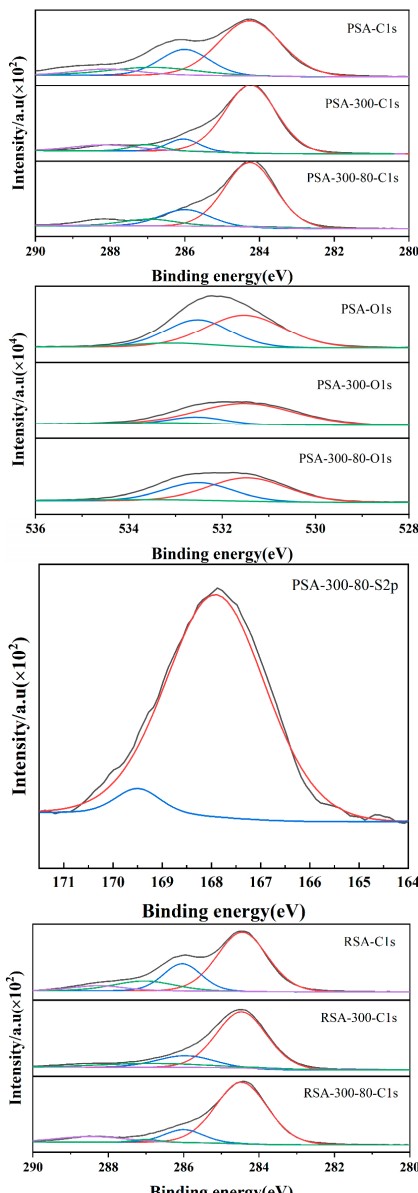

**Figure 5.** *Cont.*

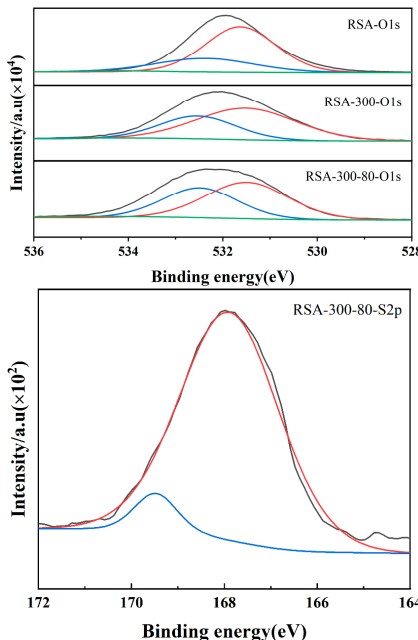

**Figure 5.** X-ray photoelectron spectroscopy (XPS) of C 1s, O 1s, and S 2p of carbon precursors and catalysts.

Table 2 presents the acid number results for the catalyst before and after activation, categorizing the acid types as $A_{Total}$, $A_{-SO3H}$, $A_{-COOH}$, and $A_{-OH}$, which represent the total, robust, medium-to-firm, and weak acids, respectively. When these findings are integrated with the hydration reaction data displayed in Table 3, a correlation emerges: the conversion of $\alpha$-pinene in the synthesis of $\alpha$-terpineol appears to be influenced by the strong acid $A_{-SO3H}$, while the selectivity towards $\alpha$-terpineol seems to be associated with the medium-strength acid $A_{-COOH}$. Compared with the unactivated rice straw carbon-based solid acid RS300-80, the strong acid $A_{-SO3H}$ in RSA-300-80 is 1.28 mmol/g, slightly lower than the 1.37 mmol/g of RS300-80. Conversely, the medium-strength acid $A_{-COOH}$ in RSA-300-80 is measured at 1.37 mmol/g, exceeding the 1.07 mmol/g found in unactivated RS300-80. RSA-300-80 demonstrates superior selectivity in the hydration reaction of $\alpha$-pinene to synthesize $\alpha$-terpineol, albeit with a lower conversion rate compared to RS300-80. Furthermore, the influences of the acid concentrations $A_{-SO3H}$ and $A_{-COOH}$ in both unactivated PS300-80 and PSA-300-80 on the conversion rate and selectivity in $\alpha$-pinene hydration are consistent with those observed for rice straw carbon-based solid acids. These findings are corroborated by the hydration reaction data presented in Table 3.

**Table 2.** Acid amount of catalysts.

| Catalysts | $A_{Total}$ (mmol/g) | $A_{-SO3H}$ (mmol/g) | $A_{-COOH}$ (mmol/g) | $A_{-OH}$ (mmol/g) |
|---|---|---|---|---|
| PS300-80 | 4.32 | 1.30 | 1.78 | 1.24 |
| PSA-300-80 | 4.20 | 1.37 | 1.05 | 1.78 |
| RS300-80 | 3.80 | 1.37 | 1.07 | 1.36 |
| RSA-300-80 | 3.93 | 1.28 | 1.37 | 1.28 |

**Table 3.** Results of catalysts' hydration reaction.

| Catalysts | Conversion (%) | Selectivity (%) | Yield (%) |
|---|---|---|---|
| PS300-80 | 85.70 | 37.35 | 32.01 |
| PSA-300-80 | 92.45 | 24.39 | 22.54 |
| RS300-80 | 89.63 | 24.42 | 21.89 |
| RSA-300-80 | 87.15 | 54.19 | 47.23 |

## 2.2. Catalytic Activity in Hydration of α-Pinene

Carbon-based solid acid catalysts are amorphous materials characterized by a plethora of acidic catalytic centers. These centers facilitate the donation of hydrogen ions to substrates while maintaining structural integrity, thus enabling catalytic processes such as hydration [38]. The vertical and cross-sectional diameters of α-pinene are measured at approximately 0.84 nm and 0.77 nm, respectively. Manual calculations reveal that the vertical diameter of α-terpineol is 1.18 nm, with a cross-sectional diameter of 1.46 nm. Additionally, the study identifies an average pore diameter of 3.58 nm for the catalysts. Consequently, these dimensions afford greater accessibility to the catalytically active centers within the catalyst, facilitating enhanced reaction rates. Furthermore, the size compatibility ensures that the target product, α-terpineol, can be readily disengaged from the catalyst interior.

As indicated in Table 3, when subjected to identical reaction conditions—temperature at 80 °C, duration of 24 h, and catalyst dosage of 0.75 g—the rice straw-based catalyst RSA-300-80 exhibits superior catalytic activity compared to its peanut shell-based counterpart PSA-300-80. Relative to the unactivated catalyst RS300-80, the newly developed RSA-300-80 catalyst demonstrates marked improvements in performance: selectivity for α-terpineol rose from 24.42% to 54.19%, and the yield increased from 21.89% to 47.23%. These enhancements are potentially attributable to chemical activation techniques that optimize the pore structure of the rice straw substrate. Concurrently, the activation process facilitates the oxidation of a fraction of the –OH groups in the biomass to –COOH. Wei et al. [39] used rice straw to prepare a carbon-based solid acid catalyst (RS300-80), and the conversion and yield of α-terpineol prepared by hydration of α-pinene were only 67.60% and 38.58%. Moreover, a comparative analysis between the catalytic performances of PSA-300-80 and the unactivated PS300-80 reveals a discernible decline in both selectivity and yield of α-terpineol. Specifically, selectivity decreases from 37.35% to 24.39%, while yield falls from 32.01% to 22.54%. This diminution may be attributed to the combined effects of $ZnCl_2$ activation and concentrated sulfuric acid sulfonation, which lead to a partial collapse of the peanut shell's pore structure [40]; consequently, a reduction in porosity is observed, a finding corroborated by the data in Table 1.

Table 4 delineates the principal products generated during the hydration reaction of α-pinene. To elucidate the catalytic efficiency of carbon-based catalysts in this context, key variables—including the reaction temperature, reaction time, and catalyst dosage—were systematically examined using RSA-300-80, a catalyst derived from rice straw. Figure 6 presents the averaged results, obtained from three independent trials, focusing solely on the conversion rates of α-pinene and the yield of the target product, α-terpineol. It should be noted that these calculations exclude the potential by-products of α-pinene hydration, such as camphene, limonene, 2-carene, and isoborneol, all of which are deemed non-toxic and harmless.

**Table 4.** The main products of the α-pinene hydration reaction.

| Number | Retention Time (min) | Name | Comparative Content (%) | Similarity (%) |
|---|---|---|---|---|
| 1 | 6.73 | (+)-Camphene | 21.44 | 89 |
| 2 | 7.12 | (-)-Camphene | 9.86 | 94 |
| 3 | 7.85 | β-Pinene | 0.19 | 94 |
| 4 | 8.64 | α-Phellandrene | 0.21 | 95 |
| 5 | 8.95 | α-Terpinene | 1.28 | 95 |
| 6 | 9.37 | Limonene | 10.39 | 89 |
| 7 | 10.16 | γ-Terpinene | 1.40 | 97 |
| 8 | 11.04 | 2-Carene | 7.62 | 94 |
| 9 | 12.18 | Fenchol | 6.48 | 94 |
| 10 | 13.44 | Isoborneol | 7.26 | 96 |
| 11 | 13.96 | 4-Terpineol | 3.90 | 96 |
| 12 | 14.71 | α-Terpineol | 29.97 | 89 |

### 2.2.1. Effect of Reaction Temperature

The influence of reaction temperature on both the conversion of α-pinene and the selectivity toward α-terpineol was examined using rice straw-based catalysts. Experimental conditions include a catalyst dosage of 0.75 g, a water-to-α-pinene molar ratio of 10:1, a reaction time of 24 h, and a solvent volume of 20 mL of acetone. As illustrated in Figure 6a, at a reaction temperature of 70 °C, the conversion of α-pinene reaches 86.05%, while the yield of α-terpineol was limited to 37.54%. Upon elevating the reaction temperature to 80 °C, the conversion of α-pinene increases marginally to 87.15%, and the yield of α-terpineol rose to 47.23%. The reason may be that when there is an acidic catalyst in the hydration reaction system, α-pinene is attacked by hydrogen ions to form carbon cations. These carbon cations undergo a ring-opening reaction or rearrangement to generate other carbon cations, which require a certain amount of energy. Therefore, when the reaction temperature is 70 °C, the yield of α-terpineol is low. When the temperature was further increased to 90 °C, the conversion of α-pinene decreased slightly, and the yield of α-terpineol decreased significantly to 35.24%. Elevated reaction temperatures accelerate the kinetic motion of acetone molecules, thereby hastening their evaporation from the system. This rapid evaporation hinders the mass transfer between the carbocationic intermediates and water, consequently diminishing the product yield. Concurrently, molecules at higher temperatures possess greater kinetic energy, increasing the propensity for undesired side reactions or driving the reaction irreversibly, both of which further compromise the yield. Overall, this study identifies 80 °C as the optimal temperature for the hydration of α-pinene when catalyzed by RSA-300-80.

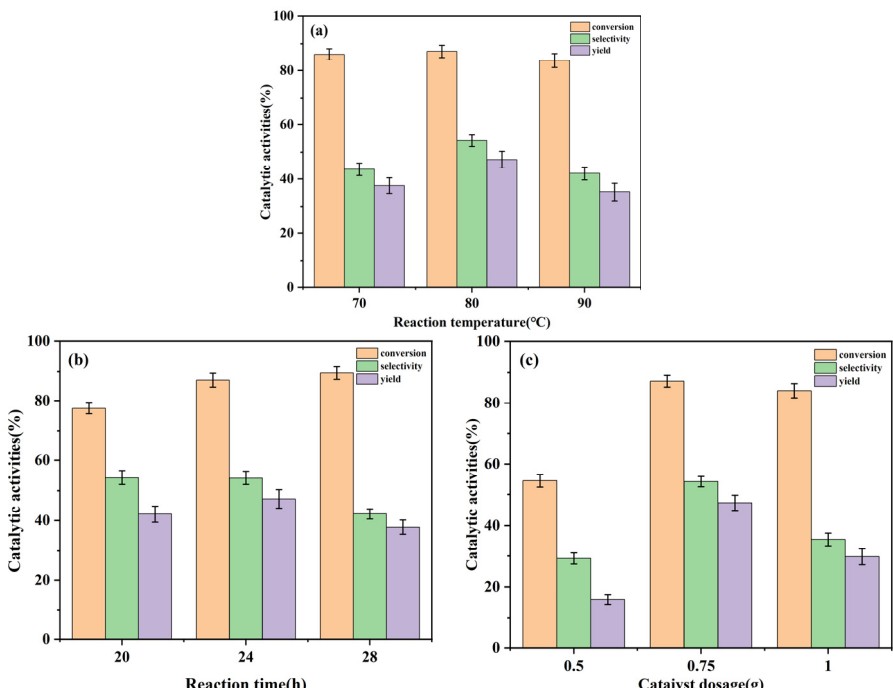

**Figure 6.** The catalytic activity of the catalysts: (**a**) effect of reaction temperature; (**b**) effect of reaction time; and (**c**) effect of catalyst input.

### 2.2.2. Effect of Reaction Time

Reaction time exerts an influence on both production costs and energy consumption; consequently, the catalytic performance was evaluated over a span of 20 to 28 h, holding other experimental parameters constant. Figure 6b displays the results. When the reaction time is extended to 24 h, both the conversion rate of α-pinene and the yield of α-terpineol show marginal increases, whereas the selectivity towards α-terpineol exhibits a moderate decline. At this 24-h juncture, the maximum yield of α-terpineol was recorded at 47.23%,

alongside an α-pinene conversion rate of 87.05%. Contrastingly, prolonging the reaction time to 28 h led to a conversion rate of α-pinene escalating to 89.51%, accompanied by a marked decrease in both the selectivity and yield of α-terpineol, which registered at 42.16% and 37.74%, respectively. This decline is attributable to isomerization reactions within the mixture; as α-terpineol accumulates, it further transforms into other by-products, thereby reducing both its selectivity and yield. These findings are congruent with extant literature [41]. Furthermore, as delineated in Scheme 1, α-terpineol is susceptible to conversion into various by-products via isomerization reactions, thereby diminishing its selectivity. For example, the transformation of α-terpineol into 2-carene proceeds through a dehydration mechanism. In the presence of a sufficient quantity of strong acid sites, such dehydration reactions are readily facilitated. Optimal results were observed at a reaction time of 24 h.

### 2.2.3. Effect of Catalyst Dosage

Employing a catalyst lowers the activation energy required for the hydration reaction of α-pinene. The reaction was conducted with varying catalyst dosages (0.50, 0.75, and 1 g). As shown in Figure 6c, the conversion of α-pinene increases, and the yield of α-terpineol increases and then decreases as the amount of catalyst increases. This trend may be attributed to the elevated number of acidic active centers in the reaction milieu with increasing catalyst quantities. Such an increase promotes the generation of additional carbon cations, thereby elevating the α-pinene conversion rate. Conversely, excessive catalyst amounts may instigate undesirable side reactions, including isomerization or degradation pathways initiated by the acidic catalyst, which consequently diminish both the selectivity and yield of the target product. In summary, supra-optimal catalyst dosages yield deleterious outcomes; the optimal catalyst dosage identified in this investigation is 0.75 g.

### 2.3. Reusability of Catalyst

To evaluate the reusability of the rice straw catalyst, three consecutive catalytic experiments were conducted under optimal conditions. The aim was to ascertain the catalyst's longevity by examining its performance in the hydration reaction over multiple cycles. Figure 7 illustrates a decline in catalytic activity correlated with an increasing number of cycles. After the third cycle, the conversion rate of α-pinene and the yield of α-terpineol diminished to 59.59% and 11.78%, respectively. This decline is potentially attributable to catalyst deactivation, site blockage, or the denaturation of active sites during the course of the reaction, which could lead either to a reduction in the number of functional active sites or to complete catalyst failure. The Fourier-transform infrared spectrum of the recycled catalyst is depicted in Figure 8. The peak in the 3080–3600 cm$^{-1}$ range corresponds to the stretching vibrations of the –OH groups. Meanwhile, the peak around 1050 cm$^{-1}$ is attributed to the asymmetric stretching of the O=S=O functional group, confirming the successful incorporation of –SO$_3$H groups into the catalyst. A discernible decrease in the peak intensity for both the –SO$_3$H and –OH groups was observed as the number of reaction cycles increased. This trend suggests that the concentration of both strong and weak acid sites positively influences the conversion of α-pinene and the yield of α-terpineol, corroborating the findings of Wei et al. [39]. The vibrational peaks appearing at approximately 2750 cm$^{-1}$ are attributed to the C–H bonds in both the C–H$_3$ and C–H$_2$ groups [19,31]. An amplification in these peak intensities is linked to the ring-opening events in the pinene hydration reaction. Further, the peaks observed at 1433, 1561, and 1687 cm$^{-1}$ are ascribed to the C=C stretching vibrations within the aromatic skeleton's absorption band. These observations suggest that the catalyst preserves its stable aromatic ring structure even after recycling procedures.

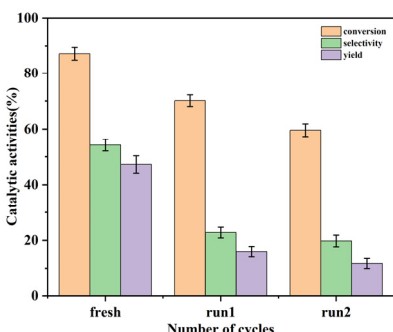

**Figure 7.** Reusability of catalyst.

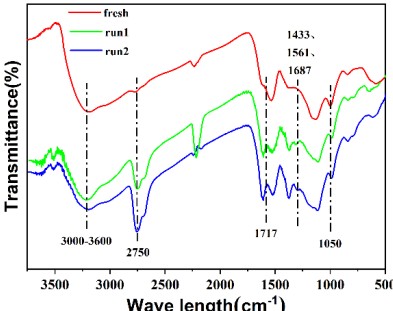

**Figure 8.** Fourier transform infrared spectroscopy (FT−IR) of the cyclic experiment.

The study revealed that the catalytic efficacy of the carbon-based solid acid catalyst for the α-pinene hydration reaction exceeded that reported by Prakoso et al., who utilized a liquid mixed acid catalyst at analogous thermal conditions. Prakoso et al. [42] used a 4:1 mass ratio of phosphoric acid and p-toluenesulfonic acid (PTSA) as catalysts to catalyze the hydration of α-pinene to prepare α-terpineol. At the optimum hydration temperature of 75 °C, the yield of α-terpineol was only 29.5%. While reusability experiments indicate room for improvement in the catalyst's activity stability, carbon-based solid acid catalysts exhibit notable advantages, including robust catalytic activity, cost-efficiency, operational simplicity, safety, and environmental compatibility. Consequently, these catalysts merit further scholarly investigation as eco-friendly alternatives.

## 3. Materials and Methods

### 3.1. Materials

Analytical-grade α-Pinene (98%) and $ZnCl_2$ (98%) were procured from Shanghai Maclean Biochemical Technology Co., Ltd. (Shanghai, China). $H_2SO_4$ (98%), acetone, and ethanol, all of analytical purity, were sourced from Chengdu Kelong Chemical Co., Ltd. (Chengdu, China). Peanut shells and rice straw were harvested manually from the experimental fields of Guangxi University, Nanning, China. Subsequent to ultrasonic cleaning to eliminate surface ash and contaminants, these biomass samples were dried at 105 °C for a duration of 8 h and stored under vacuum conditions.

### 3.2. Catalyst Preparation

Carbon-based solid acid catalysts, designated as PSA-C-S and RSA-C-S, were synthesized from the agricultural residues of peanut shells and rice straw through a multi-step process involving $ZnCl_2$ activation, carbonization, and sulfonation. In this nomenclature, "A" denotes the activation temperature, "C" represents the carbonization temperature, and "S" signifies the sulfonation temperature. The initial biomass materials—peanut shells and rice straw—were mechanically ground to a particle size below 40 mesh and subsequently stored in hermetically sealed bags.

A mass of 5 g of $ZnCl_2$ was precisely weighed and dissolved in 30 mL of deionized water. Subsequently, an equal mass of peanut shell sample was weighed and uniformly dispersed in the $ZnCl_2$ solution. This mixture was subjected to heating in an oil bath at 60 °C for a duration of 10 h. Upon cooling, the material was isolated by filtration and dried overnight to yield the preliminary solid acid PSA [19]. This PSA was then introduced into a tube furnace and heated at 300 °C for 2 h. The resultant carbonized sample underwent multiple wash cycles with a 0.5 mol/L HCl solution until the filtrate attained a neutral pH. Subsequently, the carbonized material was transferred to a beaker filled with ethanol and subjected to ultrasonic cleaning for 1 h to remove any tar formed during the carbonization stage. Finally, the sample was washed repeatedly with deionized water, filtered, and dried to obtain PSA-300. A quantity of 1 g of PSA-300 was placed in a round-bottom flask containing 10 mL of $H_2SO_4$. The mixture was then heated in an oil bath at 80 °C for a period of 3 h. Following cooling and dilution, the catalyst underwent multiple washing cycles with deionized water at 80 °C until the pH of the filtrate reached neutrality. The material was subsequently transferred to a drying oven set at 100 °C and left to dry overnight, resulting in the final product, PSA-300-80 catalyst [39]. The preparation process of the rice straw catalyst was the same as the peanut shells'.

### 3.3. Characterization of the Catalysts

The thermal stability of the sample was assessed using a simultaneous thermal analyzer (TG) under a nitrogen flow rate of 30 mL/min. The sample was subjected to a temperature range from ambient to 1000 °C at a heating rate of 10 °C/min. For morphological analysis, the samples were examined using a scanning electron microscope (SEM) supplied by Beijing Pricey Instruments Co., Ltd. Prior to SEM imaging, gold particles were sputter-coated onto each sample surface to enhance electrical conductivity. $N_2$-adsorption-desorption analysis was conducted using a Quantachrome instrument (Tallahassee, FL, USA) to examine the porosity structures of the samples at a temperature of $-196$ °C. Prior to the experiment, all samples underwent dehydration in a vacuum at 105 °C for a duration of 5 h. The specific surface area ($S_{BET}$), average pore diameter ($D_{pore}$), and total pore volume ($V_{Total}$) of the samples were calculated employing the Brunauer–Emmett–Teller (BET) equation and the Barrett–Joyner–Halenda (BJH) model, respectively. The samples were analyzed using Fourier transform infrared spectroscopy (FT-IR) on a ThermoFisher instrument (USA) to characterize their vibrational modes and functional groups. The resolution for FT-IR was set at 4 $cm^{-1}$, and each spectrum was acquired with 40 scans across a wavenumber range of 500–4000 $cm^{-1}$. For the FT-IR measurement, the samples were ground and combined with KBr powder to form pellets. Elemental analysis was performed using X-ray photoelectron spectroscopy (XPS) in a vacuum environment. The XPS was equipped with a monochromatic Al X-ray source operating at 6 μA × 3 keV, and the signal was averaged over an oval-shaped area of approximately 700 × 300 microns. The samples' total acid, sulphonic acid, and carboxylic acid amounts were determined by acid-base titration [43,44]. The specific methods were as follows:

Determination of Total Acid Value: A 0.25 g sample of the catalyst was accurately weighed using an analytical balance. Subsequently, 25 mL of a 0.05 mol/L NaOH standard solution was added to the weighed sample in a 100 mL beaker. The beaker was placed in a temperature-controlled ultrasonic cleaner at room temperature for 1 h. Following the ultrasonic cleaning, the mixture was filtered, and the filtrate was collected in a conical flask. Methyl orange was added as an indicator, and titration was conducted using a 0.05 mol/L HCl standard solution. The titration was continued, with constant agitation, until the solution in the conical flask transitioned to a red color. The volume of the consumed HCl standard solution was then recorded. The calculation formula for total acid value $A_{Total}$ is as follows:

$$A_{Total} = \frac{0.05 \times 25 - 0.05x}{m} \tag{1}$$

where $A_{Total}$ represents the total acid value, unit mmol/g; x represents the volume of HCl consumed, unit mL; and m represents the mass of the catalyst to be tested, unit g.

Determination of Sulfonic Acid Value: A total of 0.25 g of catalyst was weighed precisely using an analytical balance. Then, 25 mL of a NaCl standard solution with a concentration of 0.05 mol/L was measured in a cylinder and transferred into a 100 mL beaker. Next, the beaker was placed in a CNC ultrasonic cleaner and sonicated for 1 h at room temperature. After filtering the cleaned sample, the filtrate was collected in a conical flask. Phenolphthalein solution was added as an indicator and titrated with a 0.05 mol/L NaOH standard solution. The conical flask was then shaken and the volume of NaOH standard solution consumed when the liquid appeared slightly red and remained so for 30 s was recorded. The calculation formula for sulfonic acid value $A_{-SO_3H}$ is as follows:

$$A_{-SO_3H} = \frac{0.05x}{m} \tag{2}$$

Among them, $A_{-SO_3H}$ represents the sulfonic acid value, and the unit is mmol/g; x represents the volume of NaOH consumed in mL; and m represents the mass of the catalyst to be tested, unit g.

Determination of Carboxylic Acid Value: A total of 0.25 g of the catalyst was accurately weighed using an analytical balance. Subsequently, 25 mL of a $NaHCO_3$ standard solution with a concentration of 0.05 mol/L was measured using a graduated cylinder. This solution was transferred to a 100 mL beaker, which was then placed in a CNC ultrasonic cleaner and subjected to ultrasonication at room temperature for one hour. Following ultrasonication, the solution was filtered and the filtrate was collected in a conical flask. Methyl orange was introduced as an indicator and the solution was titrated with a standard HCl solution of 0.05 mol/L concentration. The conical flask was agitated until the solution attained a red hue, and then the volume of the standard HCl solution expended was recorded. The calculation formula for carboxylic acid value $A_{-COOH}$ is as follows:

$$A_{-SO_3H+-COOH} = \frac{0.05 \times 25 - 0.05x}{m} \tag{3}$$

$$A_{-COOH} = A_{-SO_3H+-COOH} - A_{-SO_3H} \tag{4}$$

Among them, $A_{-SO_3H+-COOH}$ represents the total acid value of sulfonic acid and carboxylic acid, and the unit is mmol/g; $A_{-COOH}$ represents the carboxylic acid value, with a unit of mmol/g; x represents the volume of HCl consumed, unit mL; and m represents the mass of the catalyst to be tested, unit g.

*3.4. Investigation of Catalytic Activity*

An exact amount of 5 mL $\alpha$-pinene, 5 mL water, 20 mL acetone, and 0.75 g of the catalyst were combined in a three-necked flask equipped with a thermometer. This assembly was subsequently submerged in an oil bath. Upon reaching the predetermined temperature, the timing of the reaction commenced. Following the stipulated reaction period, the solvent acetone was eliminated via a rotary evaporator connected to a vacuum pump. This was succeeded by high-speed centrifugation to facilitate solid–liquid separation. The resulting oily liquid was collected for analytical purposes, while the solid residue was reclaimed. To assess the durability of the catalyst under optimal hydration conditions, the solid sample underwent triple ethanol washes to remove any residual liquid product.

*3.5. Products Analysis*

Liquid products underwent analysis via a Shimadzu gas chromatograph (GC-2010, Kyoto, Japan) equipped with a flame ionization detector and a GC-2010 capillary column (30 m × 0.25 mm × 0.25 μm; Kyoto, Japan). The chromatographic oven was initially set to a temperature of 70 °C, subsequently elevated to 180 °C at a ramp rate of 2 °C/min, and finally increased to 220 °C at a rate of 15 °C/min where it was maintained for a 5-min

duration. Nitrogen served as the carrier gas, employed at a split ratio of 1:50. Both the injector and detector temperatures were stabilized at 240 °C and 280 °C, respectively.

The oil phase sample was diluted to a total volume of 10 mL using 0.25 mL of the sample and anhydrous ethanol as the diluent. Following dilution, a 1 μL aliquot of the diluted sample was injected into the gas chromatograph (GC). The conversion of α-pinene and the yield of α-terpineol were quantified through the external standard method. To facilitate this quantification, standard curves were constructed by analyzing varying concentrations of a standard solution, which comprised acetone, α-pinene, and α-terpineol. Carvone was used as the external standard [40]. The conversion of α-pinene (X) and the yield (Y) of α-terpineol were calculated using the following formulas:

$$X = \frac{m_{\alpha\text{-pinene}}}{m^0_{\alpha\text{-pinene}}} \times 100\% \tag{5}$$

$$m_{\alpha\text{-terpineol}} = \frac{A_{\alpha\text{-terpineol}} + 1 \times 10^6}{9 \times 10^7} \tag{6}$$

$$Y = \frac{m_{\alpha\text{-terpineol}}}{\frac{M_{\alpha\text{-terpineol}}}{M_{\alpha\text{-pinene}}} \times m^0_{\alpha\text{-pinene}}} \times 100\% \tag{7}$$

$$m_{\alpha\text{-pinene}} = \frac{A_{\alpha\text{-pinene}} + 2 \times 10^7}{1 \times 10^8} \tag{8}$$

$$S = \frac{Y}{X} \times 100\% \tag{9}$$

where $A_{\alpha\text{-pinene}}$ is the peak area of α-pinene, $A_{\alpha\text{-terpineol}}$ is the peak area of α-terpineol, $m^0_{\alpha\text{-pinene}}$ is the initial volume of α-pinene added, converted to α-pinene by mass (g), $m_{\alpha\text{-pinene}}$ is the mass of α-pinene consumed (g), $M_{\alpha\text{-terpineol}}$ and $M_{\alpha\text{-pinene}}$ are the relative molecular masses of α-terpineol and α-pinene, respectively, X is the conversion, Y is the yield, and S is the selectivity.

## 4. Conclusions

In summary, two catalysts—RSA-300-80 and PSA-300-80—were synthesized via zinc chloride activation followed by sulfonation using concentrated sulfuric acid, and subsequently employed in the hydration of α-pinene. Comparative analysis revealed that RSA-300-80, derived from raw material RS possessing an intrinsic porous structure, demonstrated superior catalytic efficacy in the synthesis of α-terpineol from α-pinene relative to its PSA-300-80 counterpart. Additionally, RSA-300-80 exhibited an enhanced specific surface area ($S_{BET}$) and a higher concentration of oxygen-containing functional groups (–COOH) compared to the unactivated RS300-80 sample. The specific surface areas for RS300-80 and RSA-300-80 were measured at 420.9 m$^2$/g and 527.0 m$^2$/g, respectively, while the concentrations of –COOH groups were 1.07 mmol/g and 1.37 mmol/g, respectively. This enhancement in functional groups can be attributed to the oxidation of intrinsic –OH groups to –COOH during the activation process. Conversely, when subjected to identical activation conditions, the PS300-80 and PSA-300-80 catalysts displayed contrasting trends in both their specific surface area and –COOH group concentration compared to their RS counterparts. This discrepancy arose from the collapse of certain pore structures during the carbonization and sulfonation stages of PS activation by ZnCl$_2$, resulting in a diminished specific surface area for the resultant catalyst.

The findings from the hydration experiment suggest that the elevated catalytic performance of RSA-300-80 is primarily due to two factors: a high specific surface area and an abundant presence of –COOH functional groups. The increased specific surface area facilitates reactant access to the catalyst's active sites, thereby providing a robust foundation for its superior catalytic efficacy. Concurrently, the high density of –COOH groups contributes

to the stabilization of carbocations via electrostatic interactions, effectively prolonging their lifespan. This stabilization, in turn, enhances the selectivity towards α-terpineol. Under the optimal reaction parameters of a reaction temperature of 80 °C, reaction time of 24 h, and catalyst usage of 0.75 g, RSA-300-80 catalyzed α-pinene hydration to prepare α-terpineol; the conversion of α-pinene was 87.15% and the selectivity was 54.19%; while using RS300-80, the conversion of α-pinene was 89.63%, but the selectivity of α-terpineol was only 24.42%. Furthermore, the utilization of waste biomass as the raw material for the synthesis of carbon-based solid acids was shown to offer economic and environmental advantages by minimizing solid waste residues. Employing this approach yields solid acid catalysts characterized by elevated catalytic performance, high specific surface area, and enhanced recyclability. These attributes render the catalysts highly promising for applications in α-terpineol production.

**Author Contributions:** Y.W., designed and performed the experiments, analyzed the data, wrote the paper, and assisted in software processing of the data; D.X., conceived and designed the experiments; Z.W., H.Z., M.T. and P.M. carried out SEM, BET, TG, XPS and FT-IR analysis; S.B., revised the logical representation of manuscript, and English grammatical errors, sentences, and modified structures. All authors have read and agreed to the published version of the manuscript.

**Funding:** This research was supported by the Dean Project of Guangxi Key Laboratory of Petrochemical Resource Processing and Process Intensification Technology (2021Z005) and the Innovation Project of Guangxi Graduate Education (YCSW2022010).

**Data Availability Statement:** Not applicable.

**Acknowledgments:** The authors thank Deyuan Xiong for their contribution in conceiving and designing the experiments.

**Conflicts of Interest:** The authors declare that they have no known competing financial interests or personal relationships that could have appeared to influence the work reported in this paper.

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
