# Peer review of "Structure and Catalytic Performance of Carbon-Based Solid Acids from Biomass Activated by ZnCl2"

_catalysts, doi:10.3390/catal13111436_

Round 1
Reviewer 1 Report (Previous Reviewer 2)
Comments and Suggestions for Authors
The authors have revised the manuscript according to the suggestions or comments given by the reviewers. All the issues or concerns stated by the reviewers were clearly addressed and necessary changes were done.
Comments on the Quality of English LanguageFew typographical corrections are needed.
Author Response
Please see the attachment.

Reviewer 2 Report (Previous Reviewer 3)
Comments and Suggestions for Authors
Catalysts 2664997
October 2023
Yao Wu et al
Structure and catalytic performance of Carbon-based solid acids…….
The present work is a corrected version of a preceding submission that needed a lot of modifications and up-grading.
The present work has been substantially completed and sounds better as a scientific contribution. It however still needs new data and considerations on some points before being accepted.
1/ The English use is not better than the preceding one and needs new attention. But the present reviewer will not perform this part, not list the points to be corrected.
2/ Lines 106-113: The authors introduced RESULTS in that section, whereas their experimental results have not been presented; these lines must focus on aim or hypothesis not on experimental results.
3/ Lines 118-132: here the authors describe thermal stability and thermal treatment of essentially one sample, peanut shells, whereas in the following catalytic parts, rice straw was always retained as better sample and data with peanut shells were not fully described. At least in that part results of both samples must be presented and commented.
4/ Line 197: the authors are suspecting that ZnCl2 is transformed to ZnO during thermal treatment at 300°C, under unknown atmosphere. The authors must remember that Fusion temperature of ZnCl2 is below 300°C, and that “ZnCl2 liquid phase” may diffuse in many parts of their C samples. If ZnCl2 is transformed to ZnO (the atmosphere of thermal treatment must contain oxygen), subsequent washings will not remove ZnO, and the final catalysts will contain a huge amount of Zn. But unfortunately, chemical analysis result of the samples is not given ; further, as no general XPS spectrum is presented, the reader cannot have an idea of the presence and the chemical state of Zn, or of its absence; finally, EDS data associated to SEM would also be able to see if Zn is present or not in the final catalysts samples.
More data are needed in that part.
5/ Line 207: Figure 3: As commented in my previous review, N2 adsorption -desorption experiments and analyses are unsatisfactory to reviewer. The ads-des curves are almost linear when increasing p/po, and this curve profile is unknown in sorption measurements by reviewer: the authors samples must be analysed in other labs. They also must compare their ads-des profiles to other data present in the literature such as that of P. Kaur et al, Pedosphere 33 (2023) 463-478, Influence of pyrolysis temperature on rice straw biochar properties……., https://doi.org/10.1016/j.pedsph.2022.06.046 and explain why their samples ads-des profiles are so different from classical isotherm profiles (Types I to V). A true explanation of the results is necessary. If these results are confirmed, then a new important contribution can be expected.
6/ Lines 315-325: “As shown in Figure 6a, 87.15% of α-pinene was converted as the reaction temperature was increased to 80 °C, and the yield of α-terpineol was 47.23%. When the temperature was further above 80 °C, the conversion of α-pinene was slightly reduced, and the yield of α-terpineol was significantly lower. This may be because hydrogen ions attack α-pinene to form a carbon cation when an acidic catalyst is present in the hydration reaction system. This carbon cation undergoes a ring-opening reaction or rearrangement to form other carbon cations, which requires a certain amount of energy. As a result, the conversion of α-pinene and the yield of α-terpineol are lower when the temperature is below 80 °C. Excessively high reaction temperatures also intensify the movement of acetone molecules, leading to faster acetone evaporation from the reaction system, resulting in blocked mass transfer between the carbon-cationic product and water and, therefore, inducing lower yield.”
In that section, are the authors describing what occurs at T higher than 80°C, or what is occurring at T lower than 80°C – nothing clear here. In fact, all the presented data are very close at the 3 temperatures under study, taking into account the activity and selectivity error bars.
7/ Line 396 and Figure 8: In this part, the authors try to discuss the reason for deactivation of their material during the catalytic reaction. They used FTIR data to suggest that OH groups and HSO3 ones are decreasing in efficacity. They however completely forget that huge new bands appeared at around 2750 cm-1, and such bands are possibly due to C-H stretching in compounds like “aldehyde”. If this is the case, such strong poisoning species can explain the deactivation behaviour. But in this case, how can aldehydes be formed and why they do not appear in the other products of the reaction (see Table 4).
A complete analysis of Figure 8 spectra is therefore needed.
Comments on the Quality of English LanguageNeeds of a native senior english chemist
Round 2
Reviewer 2 Report (Previous Reviewer 3)
Comments and Suggestions for Authors
Some useful corrections were performed.
Still some comments can be added to explain important deactivation during reuse and loss of selectivity.
Comments on the Quality of English LanguageCan be improved
This manuscript is a resubmission of an earlier submission. The following is a list of the peer review reports and author responses from that submission.
Round 1
Reviewer 1 Report
Comments and Suggestions for Authors
The characterization part should be more careful discussed and explained.
TGA data are provided to explain the stability of the initial material. In this case authors can also complete the data with TGA analysis for final catalysts. Especially when considering that catalyst recycle leads to the decrease of yields and the authors discuss about the collapse of the structure.
Any discussion regarding the increase of the porosity in after activation is not sustained, as no characterization (other than TGA) for starting material is presented.
The size of pores in SEM images for RSA-300-80 material seems to be lower compared to RSA-300 material. Despite that, the results of N2 adsorption-desorption suggest that surface area and pore volume increase after sulfonation step. What could be the reason for this increase? As after functionalization with SO3H group a slightly opposite effect is expected. It is also interesting that only one type of functional group containing sulfur is present in the results of XPS analysis.
FTIR spectra are small and difficult to understand.
It is recommended to perform characterization of spent catalysis and leaching test in order to explain fast deactivation of catalyst, especially when not high temperatures are required by the catalytic tests.
The authors describe the equation for calculation of selectivity in main product and discuss about the selectivity in main text. However, in the result section only Conversion and Yield are presented (in corresponding graphs). It may be important to see what the distribution of the products is. That may help further mechanistic investigations (taking into consideration that the authors discus the reason and possibility of formation of carene) or to correlate results and preparation parameters. It is therefore recommended to add this part and a proper discussion in the manuscript.
How the catalytic tests were performed at different times – were the samples taken from the same batch in time or were there different test each time. It is unlikely that the conversionwould have decreased after 24h?
Formatting issues
- Experimental part should be rewritten in third person.
- 3.4 and 3.5 parts should be reviewed and more details in catalytic tests part (regarding the weight of substrate and the amount and type of used solvent) should be also provided. These details are also missing in figures 6-7.
- It is not very clear if the acetone was used indeed as internal standard, if yes – is it an appropriate internal standard?
- Different format for cited literature are present
- The text at the end of introduction (line 111-122) represents rather the conclusion than aim of the work.
Comments on the Quality of English Language
- The manuscript need to be extensively revised for language, grammar and design.
Reviewer 2 Report
Comments and Suggestions for Authors
1. Appropriate references should be cited in the methodology section.
2. For examples, author should specify whether catalysts preparation was done using the already existing literature or not. If it is, cite the references.
3. If possible, effect of pH should also be investigated as it will affect the overall charge of the catalyst.
4. Compare the obtained results with the existing literature.
5. If possible, provide a tabulation based on the type of carbon catalyst used and outcomes observed.
6. No references were cited in the methodology section. Is entire methodology was developed by the authors? Please specify.
7. Only 28 references were used, out of which 16 were used in introduction only.
8. Authors should cite more recent literature highlighting the purpose and need of the study.
Comments on the Quality of English LanguageMinor corrections are needed, for example, representations of units throughout the manuscript is not uniform.
Reviewer 3 Report
Comments and Suggestions for Authors
Catalysts
ZnCl2 + C
Y Wu et al,
11 04 2023
General comment.
The possibility to use active C obtained from biomass waste is known and has been described hundred of times. This new contribution is therefore not quite original, although in the present case additions of ZnCl2 and Sulfate to alter the acidity is less described. The choice of the reaction is quite specific, and the reader can be unsatisfied by the fact that the other potential products, together with terpineol, are practically completely ignored. Therefore, such a situation cannot give information on the potential toxicity of the by-products, important information to deal with when an industrial process is expected.
The written draft needs an English native person to read, correct and make easier to catch the draft content, more specifically, the draft parts where the authors try to explain and discuss their results.
Looking some specific points within the draft, the authors must examine the reality of:
1/ References 14, 15, 20 and 26.
2/ In Figures 6 and 7, no idea of the reproducibility in given (error bar). As the differences in catalytic properties are not so large, the reader must understand what is real and what is within experimental un-reproducibility.
3/ In Figure 1, the presentation is not didactic: both curves must use the same “y” scale, to reveal better what are the main differences between each biomass. Further, a DTG and, if possible, a DTA trace would be useful to understand better the chemical transformations of the biomasses. The same type of curves would be welcome after each chemical modification.
4/ In Figure 2, the magnification is not given; also, the absence of quite visible porosity for the PSA samples is unexpected as the pore diameter given in Table 1, have the same type of value as the RSA samples, where porosity is fully apparent.
5/ In Figure 3, the adsorption desorption isotherms aspects are quite strange and, for reviewer, must be confirmed in other laboratory: it is practically impossible to have samples with specific surface areas between some 300 and 527 m2/g, with isotherm starting at (x=0 and y = 0). Other references showing this type of behaviour and the corresponding explanations are needed.
6/ In part 2.1.3, “Catalytic active center and acid density”, reader is curious to understand how strong, weak and intermediary acid strength sites was measured: nothing in the text, nothing in the experimental part.
7/ In Figure 4, the main lines discussed in the text must be indicated (arrow and wavelength values, in each FTIR spectrum.
8/ In Figure 5, one is expecting that the binding energy of the different deconvolution figures for C are always similar from one sample to the other and that only the intensity is changed to show the magnitude of each corresponding species. Here the BE are changed in important manner, and therefore, a simple explanation cannot be retained. Conversely, a more complex deconvolution is needed.
9/ Table 2: based on what experimental results these data were obtained/calculated? Supporting references?
10/ Line 300: the authors here speal of microporous structure of PS, whereas the BET isotherms do not show the minimum existence of such porosity. Contradiction appears therefore in the results analysis.
11/ Table 3: what were the experimental conditions that allowed the obtention of the presented results (on line 289: under the same conditions!)
12/ Line 378: "such as structural collapse”: how this comment fit the “high stability” of biomass carbons mentioned on line 61?
13/ Although the study of the reusability is welcome, the results are far from good, and this simple fact eliminates the positive conclusions of the authors.
Some details (many other details can be found within the draft, but the reviewer work is not the one of the Editor:
Without always suggesting corrections, many parts of the draft are recalled thereafter, and need deep attention of the authors.
Line 26: “α-Terpineol is of monocyclic tertiary”
Lines 43-46: “The investigation by Vital et al that involved the preparation of polydimethylsiloxane membranes filled with USY zeolite, beta zeolite or a surface modified activated carbon, were used in the hydration of α-pinene, with a maximum selectivity of 70.00%.”
Line 54: “development potential. nevertheless, »
Line 58: “catalysts are favored by is favored by researchers because”
Line 59: “It has shown that”
Line 61: “rich pores, »
Lines 63-64: “fatty acids, such as sugar, starch, palm empty fruit bunches, and jatropha seeds that often used as the raw material of catalyst.”
Line 65: “Guo Chen, etc [12]; Indika Thushari, etc [9]; Nabel A. Negm, etc »
Line 67: « used to syntheses biodiesel”
Lines 72-73: “Until now, the use of biomass carbon-based solid acid catalysts to catalyze the synthesis of α-terpineol by α-pinene is rarely reported.”
Line 76: “For the porosity defects in the biomass itself, this paper uses ZnCl2 for chemical activation to improve it.”
Line 86: “Modather F. Hussein [15] et al. used”: ref 15 starts with “Hussein” and not Modather
Line 119: “and the results showed that RSA-119 C-S had obvious advantages of recoverability and reusability” : NOT TRUE
Lines 120-122: “More importantly, under the similar reaction conditions, the prepared RSA-C-S exhibited better catalytic performance than conventional solid acids such as Amberlyst-15, Niobic acid and H-ZSM-5.”: needs reference.
Lines 128-130: “In terms of exploring the peanut shells and rice straws decompose pattern with the increase of temperature to analyze its characteristics, it has been necessary to explore about their thermal stability under”
Line 137: “among the fiber molecules”: why “fiber?”
Line 139: “and tended to be amorphous structure, and then formed the structure of polycyclic aromatic [17-20].” ?
Line 141: “of destroying the crystal zone”:?
Line 148: recall the experimental conditions in the legend.
Line 152: “pore size distribution isotherms”:?
Line 152: “As shown in Figure 3, at lower”: the authors discuss Figure 3 before discussing Figure 2
Line 166: “and provides for the subsequent loading 166 of –HSO3 groups.”?
Line 170: “and abundant porosity of different sizes.”
Line 173: “uneven arrangement of the carbon particles after the addition of the –HSO3 groups to the carbon body”: What does this mean and what are the proofs?
Line 178: “It can be found that many small holes appear on the inner”: not easy to see in the shown micrographs: the authors must indicate with arrows the places they are referring to.
Line 216: Figure 3: very strange that the profile of both catalysts shows the same irregularities in the distribution vs pore size (same small max at around 5, 8, 10 nm): What is the meaning of this?
Lines 222-224: “bands in the four main absorption zones, which are located around 1108, 1411, 1559, 1680, 1712 and 3080-3600 cm-1. However, it found the additional bands in the catalysts that did not exist in the carbon precursor, such: difficult to understand.
Line 233: “and different degrees of substitution related to C–H bending and stretching were observed [26]”? what does this mean?
Line 237: “indicate the existing aliphatic C–H was mainly denoted as -CH2-.” Origin of these -CH2?
Line 240: “stretching of SO2, indicating that the catalysts have –SO3H groups”: Difficult to understand: modification by SO4--, speaking of HSO3-, and observation of SO2 vibration: What is the chemical state of S and its degree of oxidation?
Lines 256-265: “hydration preparing α-terpineol was possibly related to the strong acid A–SO3H, and 256 the selectivity was possibly related to the medium strong acid A–COOH. Compared with the 257 unactivated rice straw carbon-based solid acid RS300-80, the number of strong acid A–SO3H 258 in RSA-300-80 was 1.28 mmol/g, slightly lower than the 1.37 mmol/g of RS300-80. How-259 ever, the number of medium-strong acid A-COOH in RSA-300-80 was 1.37 mmol/g, higher 260 than the 1.07 mmol/g of unactivated RS300-80. Based on the hydration reaction results in 261 Table 3, it can be seen that the selectivity of RSA-300-80 used for α-pinene hydration pre-262 paring α-terpineol was higher than RS300-80, but its conversion was lower than RS300-263 80. In addition, the effects of the number of A–SO3H and A–COOH of unactivated PS300-80 and 264 PSA-300-80 on the conversion and selectivity of α-pinene hydration were consistent with 265 that of rice straw carbon-based solid acid”: difficult to understand the authors points
Line 282: “of α-pinene are approximate 0.84 nm and 0.77 nm”: English
Lines 286-287: “to make reaction occur, following with target products (α-terpineol) depart from interior.” ?
Lines 299-301: “The reason for this result could be that ZnCl2 activation and concentrated sulfuric acid sulfonation make the partial microporous structure of peanut shells collapse, thus reducing the porosity of the shells, which is also confirmed by the results in Table 1.”: not demonstrated.
Line 305: “dosage on the synthesis of α-terpineol by α-pinene”: ?
Line 307: “sifting α-pinene » ?
Line 309: “terpinolene, and so on were ignored from the results.”: this is not a scientific approach.
Lines 326-329: “the Inducing lower yield. of α-pinene decreased because the reaction of α-pinene with α-terpineol was reversible exothermic, further leading to a simultaneous decrease in the conversion of α-pinene, which could be due to the inhibition of the hydration reaction of the positive reaction caused by the high temperature.”: very speculative and unsupported.
Line 331: “biomass catalyst was»: oversimplified.
Line 343: “terpineol, similar to the results of the study in references”: what references?
Lines 345-346: “they can be inferred from the side that the transformation of α-terpineol into by-products by isomerization reaction.”: difficult to understand.
Line 350: Catalyst usage or “Catalyst dosage”?
Lines 361-362: “Scheme 1 showed the reaction of hydration and isomerization were competitive, and the stability of α-terpineol was lower than some by-products, such as 2-carene.”: Scheme 1 does not show these points or comments.
Lines 381-384: “Mochida et al. Used the BEA type zeolites with 300 molar ratio SiO2/Al2O3 for catalyzing hydration of α-pinene, resulting in the conversion of α-pinene and selectivity of α-terpineol only reach 50.00% and 23.00%, respectively. Prakoso et al [28] reported that 30.69%”: the reader does not know the exact experimental conditions used by these authors: then it is difficult to compare the present results with former ones.
Line 385: “and PTSA as catalyst to”: what is PTSA?
Line 424: “obtained by simultaneous thermogravimetric” what is simultaneous thermogravimetry?
Line 433: “Brunauer–Emmet–433 Teller (BET) » : EMMETT
Comments on the Quality of English LanguageCatalysts
ZnCl2 + C
Y Wu et al,
11 04 2023
General comment.
The possibility to use active C obtained from biomass waste is known and has been described hundred of times. This new contribution is therefore not quite original, although in the present case additions of ZnCl2 and Sulfate to alter the acidity is less described. The choice of the reaction is quite specific, and the reader can be unsatisfied by the fact that the other potential products, together with terpineol, are practically completely ignored. Therefore, such a situation cannot give information on the potential toxicity of the by-products, important information to deal with when an industrial process is expected.
The written draft needs an English native person to read, correct and make easier to catch the draft content, more specifically, the draft parts where the authors try to explain and discuss their results.
Looking some specific points within the draft, the authors must examine the reality of:
1/ References 14, 15, 20 and 26.
2/ In Figures 6 and 7, no idea of the reproducibility in given (error bar). As the differences in catalytic properties are not so large, the reader must understand what is real and what is within experimental un-reproducibility.
3/ In Figure 1, the presentation is not didactic: both curves must use the same “y” scale, to reveal better what are the main differences between each biomass. Further, a DTG and, if possible, a DTA trace would be useful to understand better the chemical transformations of the biomasses. The same type of curves would be welcome after each chemical modification.
4/ In Figure 2, the magnification is not given; also, the absence of quite visible porosity for the PSA samples is unexpected as the pore diameter given in Table 1, have the same type of value as the RSA samples, where porosity is fully apparent.
5/ In Figure 3, the adsorption desorption isotherms aspects are quite strange and, for reviewer, must be confirmed in other laboratory: it is practically impossible to have samples with specific surface areas between some 300 and 527 m2/g, with isotherm starting at (x=0 and y = 0). Other references showing this type of behaviour and the corresponding explanations are needed.
6/ In part 2.1.3, “Catalytic active center and acid density”, reader is curious to understand how strong, weak and intermediary acid strength sites was measured: nothing in the text, nothing in the experimental part.
7/ In Figure 4, the main lines discussed in the text must be indicated (arrow and wavelength values, in each FTIR spectrum.
8/ In Figure 5, one is expecting that the binding energy of the different deconvolution figures for C are always similar from one sample to the other and that only the intensity is changed to show the magnitude of each corresponding species. Here the BE are changed in important manner, and therefore, a simple explanation cannot be retained. Conversely, a more complex deconvolution is needed.
9/ Table 2: based on what experimental results these data were obtained/calculated? Supporting references?
10/ Line 300: the authors here speal of microporous structure of PS, whereas the BET isotherms do not show the minimum existence of such porosity. Contradiction appears therefore in the results analysis.
11/ Table 3: what were the experimental conditions that allowed the obtention of the presented results (on line 289: under the same conditions!)
12/ Line 378: "such as structural collapse”: how this comment fit the “high stability” of biomass carbons mentioned on line 61?
13/ Although the study of the reusability is welcome, the results are far from good, and this simple fact eliminates the positive conclusions of the authors.
Some details (many other details can be found within the draft, but the reviewer work is not the one of the Editor:
Without always suggesting corrections, many parts of the draft are recalled thereafter, and need deep attention of the authors.
Line 26: “α-Terpineol is of monocyclic tertiary”
Lines 43-46: “The investigation by Vital et al that involved the preparation of polydimethylsiloxane membranes filled with USY zeolite, beta zeolite or a surface modified activated carbon, were used in the hydration of α-pinene, with a maximum selectivity of 70.00%.”
Line 54: “development potential. nevertheless, »
Line 58: “catalysts are favored by is favored by researchers because”
Line 59: “It has shown that”
Line 61: “rich pores, »
Lines 63-64: “fatty acids, such as sugar, starch, palm empty fruit bunches, and jatropha seeds that often used as the raw material of catalyst.”
Line 65: “Guo Chen, etc [12]; Indika Thushari, etc [9]; Nabel A. Negm, etc »
Line 67: « used to syntheses biodiesel”
Lines 72-73: “Until now, the use of biomass carbon-based solid acid catalysts to catalyze the synthesis of α-terpineol by α-pinene is rarely reported.”
Line 76: “For the porosity defects in the biomass itself, this paper uses ZnCl2 for chemical activation to improve it.”
Line 86: “Modather F. Hussein [15] et al. used”: ref 15 starts with “Hussein” and not Modather
Line 119: “and the results showed that RSA-119 C-S had obvious advantages of recoverability and reusability” : NOT TRUE
Lines 120-122: “More importantly, under the similar reaction conditions, the prepared RSA-C-S exhibited better catalytic performance than conventional solid acids such as Amberlyst-15, Niobic acid and H-ZSM-5.”: needs reference.
Lines 128-130: “In terms of exploring the peanut shells and rice straws decompose pattern with the increase of temperature to analyze its characteristics, it has been necessary to explore about their thermal stability under”
Line 137: “among the fiber molecules”: why “fiber?”
Line 139: “and tended to be amorphous structure, and then formed the structure of polycyclic aromatic [17-20].” ?
Line 141: “of destroying the crystal zone”:?
Line 148: recall the experimental conditions in the legend.
Line 152: “pore size distribution isotherms”:?
Line 152: “As shown in Figure 3, at lower”: the authors discuss Figure 3 before discussing Figure 2
Line 166: “and provides for the subsequent loading 166 of –HSO3 groups.”?
Line 170: “and abundant porosity of different sizes.”
Line 173: “uneven arrangement of the carbon particles after the addition of the –HSO3 groups to the carbon body”: What does this mean and what are the proofs?
Line 178: “It can be found that many small holes appear on the inner”: not easy to see in the shown micrographs: the authors must indicate with arrows the places they are referring to.
Line 216: Figure 3: very strange that the profile of both catalysts shows the same irregularities in the distribution vs pore size (same small max at around 5, 8, 10 nm): What is the meaning of this?
Lines 222-224: “bands in the four main absorption zones, which are located around 1108, 1411, 1559, 1680, 1712 and 3080-3600 cm-1. However, it found the additional bands in the catalysts that did not exist in the carbon precursor, such: difficult to understand.
Line 233: “and different degrees of substitution related to C–H bending and stretching were observed [26]”? what does this mean?
Line 237: “indicate the existing aliphatic C–H was mainly denoted as -CH2-.” Origin of these -CH2?
Line 240: “stretching of SO2, indicating that the catalysts have –SO3H groups”: Difficult to understand: modification by SO4--, speaking of HSO3-, and observation of SO2 vibration: What is the chemical state of S and its degree of oxidation?
Lines 256-265: “hydration preparing α-terpineol was possibly related to the strong acid A–SO3H, and 256 the selectivity was possibly related to the medium strong acid A–COOH. Compared with the 257 unactivated rice straw carbon-based solid acid RS300-80, the number of strong acid A–SO3H 258 in RSA-300-80 was 1.28 mmol/g, slightly lower than the 1.37 mmol/g of RS300-80. How-259 ever, the number of medium-strong acid A-COOH in RSA-300-80 was 1.37 mmol/g, higher 260 than the 1.07 mmol/g of unactivated RS300-80. Based on the hydration reaction results in 261 Table 3, it can be seen that the selectivity of RSA-300-80 used for α-pinene hydration pre-262 paring α-terpineol was higher than RS300-80, but its conversion was lower than RS300-263 80. In addition, the effects of the number of A–SO3H and A–COOH of unactivated PS300-80 and 264 PSA-300-80 on the conversion and selectivity of α-pinene hydration were consistent with 265 that of rice straw carbon-based solid acid”: difficult to understand the authors points
Line 282: “of α-pinene are approximate 0.84 nm and 0.77 nm”: English
Lines 286-287: “to make reaction occur, following with target products (α-terpineol) depart from interior.” ?
Lines 299-301: “The reason for this result could be that ZnCl2 activation and concentrated sulfuric acid sulfonation make the partial microporous structure of peanut shells collapse, thus reducing the porosity of the shells, which is also confirmed by the results in Table 1.”: not demonstrated.
Line 305: “dosage on the synthesis of α-terpineol by α-pinene”: ?
Line 307: “sifting α-pinene » ?
Line 309: “terpinolene, and so on were ignored from the results.”: this is not a scientific approach.
Lines 326-329: “the Inducing lower yield. of α-pinene decreased because the reaction of α-pinene with α-terpineol was reversible exothermic, further leading to a simultaneous decrease in the conversion of α-pinene, which could be due to the inhibition of the hydration reaction of the positive reaction caused by the high temperature.”: very speculative and unsupported.
Line 331: “biomass catalyst was»: oversimplified.
Line 343: “terpineol, similar to the results of the study in references”: what references?
Lines 345-346: “they can be inferred from the side that the transformation of α-terpineol into by-products by isomerization reaction.”: difficult to understand.
Line 350: Catalyst usage or “Catalyst dosage”?
Lines 361-362: “Scheme 1 showed the reaction of hydration and isomerization were competitive, and the stability of α-terpineol was lower than some by-products, such as 2-carene.”: Scheme 1 does not show these points or comments.
Lines 381-384: “Mochida et al. Used the BEA type zeolites with 300 molar ratio SiO2/Al2O3 for catalyzing hydration of α-pinene, resulting in the conversion of α-pinene and selectivity of α-terpineol only reach 50.00% and 23.00%, respectively. Prakoso et al [28] reported that 30.69%”: the reader does not know the exact experimental conditions used by these authors: then it is difficult to compare the present results with former ones.
Line 385: “and PTSA as catalyst to”: what is PTSA?
Line 424: “obtained by simultaneous thermogravimetric” what is simultaneous thermogravimetry?
Line 433: “Brunauer–Emmet–433 Teller (BET) » : EMMETT